# The Antiaging Activities of Phytochemicals in Dark-Colored Plant Foods: Involvement of the Autophagy- and Apoptosis-Associated Pathways

**DOI:** 10.3390/ijms231911038

**Published:** 2022-09-20

**Authors:** Mengliu Luo, Meiqing Mai, Wanhan Song, Qianhua Yuan, Xiaoling Feng, Enqin Xia, Honghui Guo

**Affiliations:** 1Department of Nutrition, School of Public Health, Guangdong Medical University, Dongguan 523808, China; 2Dongguan Key Laboratory of Environmental Medicine, School of Public Health, Guangdong Medical University, Dongguan 523808, China

**Keywords:** anti-aging, phytochemicals, autophagy, apoptosis

## Abstract

In the last two decades, human life expectancy has increased by about 10 years, but this has not been accompanied by a corresponding increase in healthy lifespan. Aging is associated with a wide range of human disorders, including cancer, diabetes, and cardiovascular and neurodegenerative diseases. Delaying the aging of organs or tissues and improving the physiological functions of the elderly can reduce the risk of aging-related diseases. Autophagy and apoptosis are crucial mechanisms for cell survival and tissue homeostasis, and may also be primary aging-regulatory pathways. Recent epidemiological studies have shown that eating more colorful plant foods could increase life expectancy. Several representative phytochemicals in dark-colored plant foods such as quercetin, catechin, curcumin, anthocyanins, and lycopene have apparent antiaging potential. Nevertheless, the antiaging signaling pathways of the phytochemicals from dark-colored plant foods remain elusive. In the present review, we summarized autophagy- and apoptosis-associated targeting pathways of those phytochemicals and discussed the core targets involved in the antiaging effects. Further clinical evaluation and exploitation of phytochemicals as antiaging agents are needed to develop novel antiaging therapeutics for preventing age-related diseases and improving a healthy lifespan.

## 1. Introduction

Life expectancy is a significant indicator for estimating an individual’s quality of life. The United Nations estimate for the global average life expectancy increased from 66.8 in 2000 to 73.3 in 2019, indicating an increase in the elderly population. The wasting and weakening of the tissues that accompany senescence are associated with an array of age-related chronic diseases, such as cardiovascular diseases, chronic obstructive pulmonary disease, type 2 diabetes mellitus (T2DM), osteoporosis, and cancers [1,2,3]. In recent years, gerontology research has provided plenty of evidence that aging can be modulated by the rate and quality of aging [4]. Thus, exploring how to scientifically and effectively resist aging has become a top priority.

Healthy aging and human longevity are intricate phenotypes affected by environmental factors, such as physical exercise, diet, and psychosocial circumstances. A great number of studies on dietary intervention have illustrated that a nutritious diet and caloric restriction have an essential role in healthy aging [5,6,7]. Diets rich in whole grains, fruits, and vegetables can promote health, delay aging, and reduce the incidence of aging-related chronic diseases [8,9]. The reason is that these plant foods contain a high amount of fiber, vitamins, and minerals, but are low in calories. Higher fiber intake is associated with lower mortality risk for those with T2DM and cardiovascular diseases. Additionally, many bioactive compounds called phytonutrients or phytochemicals, which are derived from grains, fruits, and vegetables, especially dark-colored varieties, play a significant role in human disease prevention. These phytochemicals have powerful antioxidant activities and multiple health benefits, such as anti-inflammation, anti-atherogenicity, anticancer, repairing the nervous system, and delaying aging [10,11]. Among those phytochemicals, curcumin, anthocyanins, lycopene, catechin, and quercetin are of great interest due to their wide distribution in plant foods. These phytochemicals have potent power for antiaging properties in many models of aging organisms, including yeast, *Drosophila melanogaster*, *Caenorhabditis elegans*, and rodents [5,12,13].

Aging is associated with an aberrant function of multiple signaling pathways and a host of factors that maintain cellular homeostasis. A growing body of evidence has shown that abnormal apoptosis and autophagy are hallmarks of aging (Figure 1). The process of cellular senescence is usually accompanied by a decrease in the level of autophagy, which reduces the degradation ability of damaged organelles and proteins; a decrease in the level of autophagy can accelerate the aging process. Moderate activation of autophagy has antiaging effects and prolongs lifespan in many model organisms by eliminating defective cellular molecules, reducing oncogenic transformation, or increasing hormesis. Apoptosis is a genetically controlled and evolutionarily conserved form of cell death that is evident in cells exposed to stressful stimuli or injury, thereby preventing the proliferation of potentially dangerous cells.

In this review, we focused on the relationship between phytochemicals, which are richly available in dark-colored foods, and the aging process. Experimental studies in vivo or in vitro, meta-analyses, randomized controlled trials, and reviews published between January 2001 and August 2022 were sought through PubMed and Science Direct, using the following search terms: (‘phytochemicals’ or ‘autophagy’ or ‘apoptosis’) and (‘life expectancy’ or ‘healthy aging’). By searching the relevant literature, we comprehensively reviewed the literature about the antiaging effects of phytochemicals in recent years and summarized the involved antiaging signaling pathways. Additionally, we highlighted useful information for the potential application of phytochemicals to delay human aging and improve the health status of the elderly.

## 2. Role of Autophagy and Apoptosis in Aging

### 2.1. Autophagy

Autophagy is a catabolism process through utilizes lysosomes to degrade damaged, denatured, and aged proteins and organelles in cells. It involves the sequestration and transport of intracellular material to the lysosomes for degradation. According to the differences in the substrate transport processes, the autophagy processes can be divided into macroautophagy, chaperone-mediated autophagy, and microautophagy [14]. Macroautophagy is representative and the principal type of autophagy. It is an intracellular mechanism for degrading long-living proteins, preventing the accumulation of toxic substances, and promoting longevity [15,16]. In a physiological state, the three autophagy pathways cooperate with each other. When one of the autophagy pathways is blocked, the other autophagy pathway is activated and is highly expressed to complete the removal of abnormal intracellular proteins, enabling cells to maintain metabolic balance. They remove those proteins that are damaged or misfolded and the damaged organelles to maintain the normal operation state of the cells. The aggregation and accumulation of abnormal proteins can produce cytotoxicity, which affects the function of cells and causes a series of diseases.

Several relative genes are responsible for autophagy during the coordinated actions which could result in the regulation phase of autophagy (Figure 1). More often than not, autophagy is regulated by a core autophagy-related gene (Atg). Recently, over 30 types of Atg have been identified from many organisms, ranging from yeasts to mammals [16]. Atg1, also regarded as uncoordinated-51-like kinase 1 (ULK1), is a serine-threonine kinase and plays a crucial function in the initial activation of autophagy. Atg1 forms a complex protein together with autophagy proteins Atg13, FAK family kinase interacting protein of 200 kDa (FIP200), and Atg101 to regulate the initiation of autophagy [17]. Moreover, Atg1 protects tumor cells from enzalutamide-induced apoptosis through increasing autophagy flux [18]. The mammalian ortholog of yeast Atg6 is Beclin1, which plays a central role in autophagy and constitutes a molecular platform for the adjustment of autophagosome formation [19]. As an essential tumor suppressor, it modulates the initiation and regulation of autophagy. It interacts with several cofactors to induce autophagy [20]. It is known to interact with the B-cell lymphoma 2 (Bcl-2) family of antiapoptotic proteins, and these interactions inhibit Beclin1-mediated autophagy. Pharmacological or genetic manipulations that stimulate autophagy has been demonstrated to increase lifespan in model organisms. Targeted transcription factor EB (TFEB) is a crucial regulator of the autophagy–lysosomal pathway. TFEB-mediated autophagy promotes the scavenging of damaged mitochondria and excess reactive oxygen species (ROS) [21].

In the aging process, autophagy plays an essential role as an accelerator of cell death or survival. Moreover, as a cellular protection mechanism, autophagy can eliminate the damaged organelles, but overactive autophagy can also give rise to the death of cells [16,22].

### 2.2. Apoptosis

Apoptosis is a highly regulated, active process of cell death involved in the development, homeostasis, and aging. Apoptosis can be triggered in a cell through either the caspase-mediated extrinsic or intrinsic pathways (Figure 1). The intrinsic apoptotic pathway, which is activated by several intracellular stimuli, including DNA damage and oxidative stress, is composed of procaspase-9, apoptotic protease-activating factor (Apaf-1), and cytochrome c. The intrinsic pathway is intently regulated by the Bcl-2 family of intracellular proteins, whereas the extrinsic pathway of apoptosis is initiated by the binding of death ligands. These two pathways converge on caspases-3, caspases-6, or caspases-7 allowing the disruption of DNA and cellular components and inducing the typical morphological changes in apoptosis [23].

An essential characteristic of apoptosis is the release of cytochrome c from mitochondria and regulated by a balance between proapoptotic and antiapoptotic proteins of the Bcl-2 family. Antiapoptotic proteins regulate apoptosis by blocking the mitochondrial release of cytochrome c, while proapoptotic proteins act by promoting this release. The Bcl-2 family inhibits apoptosis by reducing caspase activation such as caspase-3 and caspase-8. Bax protein, a member of the Bcl-2 family of proteins, is a regulator of apoptosis and promotes apoptosis [24]. Furthermore, it has been reported that the failure of stress-induced downregulation of the Bcl-2 family in human fibroblasts contributed to the resistance of senescent fibroblasts to apoptosis [25]. Therefore, this suggests apoptosis of its caspase machinery is a booster to the aging process.

## 3. Phytochemicals in Dark-Colored Plant Foods and Their Antiaging Actions

Phytochemicals are abundant in dark-colored grains, vegetables, and fruits, and they can act on distinct antiaging signaling pathways targets, by activating autophagy, and/or by inhibiting apoptosis to delay aging. The apoptosis and the autophagy process involve numerous signaling pathways, such as insulin/insulin-like growth factor (IGF)-1, mammalian target of rapamycin (mTOR), nuclear factor kappa B (NF-κB), adenosine monophosphate-activated (AMPK), mitogen-activated protein kinase (MAPK), Sirtuin-Forkhead box o (Sirt-FOXO), the serine-/threonine-specific protein kinase (AKT), and phosphatidylinositol 3-kinase (PI3K) signaling pathways (Figure 2). These signaling transduction pathways regulate the intracellular levels of cyclic adenosine monophosphate (cAMP) and nicotinamide adenine dinucleotide (NAD^+^), and they form sophisticated networks with each other related to longevity and aging.

### 3.1. Curcumin

Curcumin is an acidic polyphenolic compound derived from rhizomes of plants such as turmeric, mustard, and curry. It is a promising antiaging compound, which is safe, inexpensive, and easily available in the diet. During food digestion, curcumin is poorly absorbed by intestinal cells due to low aqueous solubility and stability, but it can be rapidly metabolized in the liver. The intake of curcumin is about 80–200 mg/d in adults, and extremely high daily doses of curcumin intake (12 g/d) are harmless to humans [10].

Aging is often accompanied by a decline or diminished function of multiple organs and tissues, especially the nervous and cardiovascular systems. In recent years, several clinical investigations demonstrated that curcumin has a therapeutic ability against age-related chronic disorders, such as pulmonary, neoplastic, cardiovascular, T2DM, and neurological sickness [26,27,28,29,30]. Recently, investigations about curcumin on aging and age-associated diseases have indicated that curcumin and its metabolites prolong the mean lifespan of some aging model organisms such as *Drosophila melanogaster*, *Caenorhabditis elegans*, yeast, and rodents [5,27]. Lifespan extension by curcumin in *Caenorhabditis elegans* was attributed to its antioxidative properties [31].

Oxidative stress and exacerbated inflammatory responses can significantly promote aging. Curcumin supplementation (80 mg/d) in healthy middle-aged people (40–60 years old) delayed cellular aging, reduced aging-related oxidative stress, and had a preventive effect on age-related decline in vascular function [32]. In a study on the effects of curcumin long-term administration (72 weeks) on vascular dysfunction and chronic inflammation in high-fat-diet-fed mice, Takano et al. found that curcumin reduced the accumulation of senescent cells in the aorta of aged mice [33]. In a clinical trial evaluating the efficacy of curcumin in delaying the development of T2DM in people with prediabetes, no one in the curcumin-treated group (1.5 g/d) was diagnosed with T2DM after 9 months of treatment [34].

In fact, curcumin can interact with multiple molecular targets and modulate their activities including enzymes, inflammatory cytokines, transcription factors, growth factors, hormones, receptors, adipokines, and diverse signaling cascades.

MAPKs are protein kinases involved in cell survival, such as cell proliferation, differentiation, migration, and apoptosis. P38 belongs to the MAPK family, along with ERK1/2, ERK5, and c-Jun amino-terminal kinase (JNK). Cao et al. found that curcumin could prevent atherosclerosis by downregulating the activity of NF-κB and p38 MAPK signaling pathways, and reducing the expression of matrix metalloproteinases-9 (MMP-9) in macrophages induced oxidized low-density lipoprotein [35]. Curcumin could also protect the brain of mice with oxidative damage by inhibiting the phosphorylation of NF-κB/JNK, downregulating the expression of proapoptotic proteins Bax (Bcl-2-associated X protein), caspase-3 and upregulating the expression of antiapoptotic protein Bcl-2 [36]. Liang et al. studied autophagosome formation and axonal transport in an Alzheimer’s disease cell model and found that curcumin significantly upregulated the expression of autophagy-related proteins Beclin1, Atg5, and Atg16L1, enhanced autophagic flux, and promoted autophagosome–lysosome fusion, thereby exerting neuroprotective effects [37]. Curcumin can inhibit PI3K/AKT/mTOR downstream phosphorylation of p70 S6 kinase 1 (S6K1) and eukaryotic initiation factor 4E (eIF4E). It regulates the activity of Atgl/ULK and other autophagic-related molecules and eventually performs a negative function [38]. Curcumin could efficiently reduce the accumulation of A53T α-synuclein through downregulation of the mTOR/p70S6K signaling and recovery of macroautophagy in A53T α-synuclein cell model of Parkinson’s disease, indicating that curcumin could be a candidate neuroprotective agent by inducing macroautophagy [39]. Additionally, curcumin could prevent Alzheimer’s disease progression by reducing β-amyloid (Aβ) production and induce autophagy by downregulating the PI3K/AKT/mTOR signaling pathway in amyloid β precursor protein (APP)/presenilin 1 (PS1) double transgenic mice [40]. On the other hand, curcumin could induce autophagy by activating the AMPK signaling pathway or inhibiting the PI3K/AKT/mTOR signaling pathway [41]. Curcumin could also inhibit the NF-κB signaling pathway and prevent inflammation by reducing the number of inflammatory factors in serum, such as tumor necrosis factor-α (TNF-α), interleukin-6 (IL-6), and IL-8 [42].

Curcumin has various functions, especially antioxidant and anti-inflammatory effects, which is the main basis for its antiaging action. It promotes apoptosis and induces autophagy by downregulating the NF-κB, PI3K/AKT/mTOR, and MAPK signaling pathways, thereby exerting a protective effect on the nervous system and cardiovascular system.

### 3.2. Anthocyanins

Anthocyanins are the major water-soluble pigment in dark-colored plants, such as blueberries, strawberries, purple corn, and purple sweet potatoes [43]. Blue, red, and purple anthocyanins extracts have traditionally been used as dyes and food colorants [11]. Anthocyanins are classified, based on the number of hydroxyl and methoxy groups attached to the brink, into six types: cyanidin, delphinidin, pelargonidin, peonidin, malvidin, and petunidin [44]. Anthocyanins are absorbed through the stomach wall with an absorption rate of 10–22%, depending on their chemical structure, and their bioavailability is about 0.26–1.8% [45]. Recent studies of Australian, Chinese, and European populations have indicated that the average daily intake of anthocyanins ranges from 19.8 to 64.9 mg/d [46,47,48].

In addition to the various functions in plants, anthocyanins have an array of health benefits for humans. To date, anthocyanins have been confirmed to be effective in preventing various diseases based on epidemiological and clinical studies [49]. Previous human studies have confirmed that owing to their antioxidative and anti-inflammatory capacity, anthocyanin-rich foods, or anthocyanins extracts decreased the risk of obesity-related chronic diseases, such as cardiovascular diseases and T2DM [49,50,51]. A recent meta-analysis of cohort studies demonstrated that the risk of T2DM was reduced by 5% with each 7.5 mg/d increment of anthocyanins intake [50]. Dietary anthocyanin-rich bilberry extract reduced blood glucose levels and enhanced insulin sensitivity in T2DM mice models [52].

In addition, anthocyanins are a natural neuroprotective agent. Anthocyanins have been associated with multiple benefits that are pertinent to neurodegeneration such as enhanced neuronal signaling and resilience [53,54]. Preclinical research has demonstrated that red berries supplementation exerts both neuroprotective functional and mechanistic effects, such as reversal of age-related decrements in cognition, enhancements of neuronal resilience, endothelial protection, and cerebral blood flow [53]. Anthocyanins in purple sweet potato alleviated ischemic stroke in rats by reducing apoptosis-inducing factor levels and other antioxidant mechanisms [55].

Neurodegenerative diseases are often accompanied by the accumulation of abnormal intracellular proteins. By inducing the autophagy pathway, enhancing the various stages of autophagy can alleviate clinical symptoms of neurodegenerative diseases. Recently, it was reported that anthocyanins from purple sweet potato improved cognitive deficits in high-fat-diet-fed mice by enhancing autophagy [56]. Likewise, anthocyanins from purple sweet potato increased the antioxidant enzyme activity and gene expressions, inhibited the mTOR pathway and activated the autophagy pathway, thus extending the lifespan of *Drosophila melanogaster* [57].

The main biological roles of the IGF-1 signaling pathway include regulation of cell metabolism, growth, proliferation, and apoptosis in multiorgan systems. In a rat model, oral administration of black rice anthocyanins reduced cardiomyocyte apoptosis, thereby protecting the cardiac function in rats. It has been reported that the effects of anthocyanins on preventing apoptosis and cardiac dysfunction in diabetic rats are mediated by activating IGF-1 receptors and signaling pathways [58]. In addition, purple wheat is high in anthocyanins, and the effects of purple wheat on prolonging the lifespan of *Caenorhabditis elegans* are associated with the IGF-1 signaling pathway [59]. AKT-1 and DAF-16/FOXO are vital factors in the IGF-1 signaling pathway, which is associated with longevity in *Caenorhabditis elegans*. Anthocyanins extract from cherry played a crucial role in the IGF-1 signaling pathway directly or indirectly affecting the activity of DAF-16 transcription factors [60].

Mechanistically, recent studies have found that anthocyanins regulate the AMPK signaling pathway to perform its biological function. It has been explained that anthocyanins can activate the AMPK signaling pathway, which is involved in the regulation of energy homeostasis and influenced the activity of many enzymes. Anthocyanin-rich extract from black soybean improved diabetic nephropathy by activating the AMPK signaling pathway to inhibit apoptosis and oxidative stress [61]. Anthocyanins also inhibited the activation of the apoptosis-signal-regulated kinase 1 (ASK1)-JNK/p38 pathway. Additionally, mulberry anthocyanins activated MAPK transcription factors and their downstream targets that are related to oxidative stress and extended the life of *Caenorhabditis elegans* by activating p38 [62].

Anthocyanins are safe, have low–no toxicity, and have certain nutritional and pharmacological effects. Recent evidence suggests that anthocyanins with the capacity to modulate autophagy and apoptosis signaling pathways in aging-related diseases.

### 3.3. Quercetin

Quercetin ubiquitously exists in vegetables and fruits such as onions, asparagus, apples, cherries, and berries. Most quercetin is present in plants as hydrophilic glycosides that are not easily absorbed directly. The intake of quercetin for healthy young males in China is about 18 mg/d. The median intake of quercetin in Japan was 15.5 mg/d. The average intake of quercetin was about 9.75 mg/d for U.S. adults [63].

The secondary metabolites of flavonoids in fruits and vegetables are considered to be one of the bioactive compounds with beneficial cardiovascular effects. Among them, quercetin is the most widely distributed flavonoid in food. Dietary intake of quercetin is associated with a reduced risk of cardiovascular disease. Quercetin intake at 150–730 mg/d for 4–10 weeks had antihypertensive effects [64]. It altered the values of various blood pressure regulators such as vascular compliance, peripheral vascular resistance, and total blood volume through its anti-inflammatory and antioxidant effects [64]. Quercetin restored the transmembrane potential altered by advanced glycation end products (AGEs) in vitro, thus showing antidiabetic potential in the late stages of diabetes [65].

Consumption of quercetin-rich vegetables was associated with a decreased risk of various types of cancers, including breast, lung, nasopharyngeal, kidney, colorectal, prostate, pancreatic, and ovarian cancers [66]. Quercetin enhanced the intracellular Ca^2+^ level leading to disruption of mitochondrial membrane potential, provoking cytochrome c release, and activating caspase-3/7 [67]. In addition, quercetin also induced apoptosis via the mitochondrial pathway by activating the caspase cascade [68]. Quercetin can induce apoptosis, induce cell cycle arrest, and inhibit the mitotic process by regulating molecular pathways such as cyclin, pro-apoptosis, PI3K/AKT, and MAPK to achieve an antitumor effect. In rats of CCl_4_-induced liver fibrosis, quercetin exerted anti-inflammatory, antiapoptotic, and antifibrotic effects by regulating NF-κB/IκBα, p38 MAPK, and Bcl-2/Bax signaling pathways [69]. When quercetin was administered to rats at 50 mg/kg body weight, it attenuated neuronal autophagy and apoptosis in a rat model of traumatic brain injury by activating the PI3K/Akt signaling pathway [70].

Notably, quercetin induces apoptosis and inhibits the migration of bladder cancer cells via activation of the AMPK pathway [71]. Activation of AMPK can induce p53-mediated cell cycle arrest, thereby inhibiting tumor cell growth and downregulating mTORC1 activity in rodents. In addition, quercetin modulated the AMPK/Sirt1/NF-κB signaling pathway to inhibit inflammatory and oxidative stress responses in the carotid artery of high-fat-diet-induced atherosclerotic rats [72].

Regulation of the IGF-1 signaling pathway is considered to underlie interactions between longevity, diet, and metabolism in a wide variety of organisms [73]. Quercetin feeding prolonged lifespan, suppressed age-related motility retardation, improved motility recovery after heat stress, and decreased the production of both intercellular and mitochondrial reactive oxygen species in *Caenorhabditis elegans*, and this process was regulated by the IGF-1 signaling pathway and the transcription factor DAF-16 [74].

As a ubiquitous flavonoid, quercetin is a safe dietary supplement due to its broad health-promoting effects in humans. It improves atherosclerosis and inhibits cancer development mainly by regulating IGF-1 and AMPK signaling pathways.

### 3.4. Catechin

Catechin is a class of highly active polyphenols present in a variety of foods such as tea, berries, cocoa beans, legumes, and nuts. Major catechin include (-)-epigallocatechin-3-gallate (EGCG), (-)-epigallocatechin (EGC), (-)-epicatechin-3-gallate (ECG), and (-)-epicatechin (EC) [75]. Catechin is the most active polyphenolic compound in tea, accounting for more than 50% of the total polyphenols’ contents in green tea. Only a small fraction of tea catechin present in the intestinal tract after drinking tea can be absorbed into the portal vein. Approximately 1.68% of ingested catechin were present in the plasma, urine, and feces, after black tea ingestion over 6 h [76]. Catechin represents strong antioxidant properties that help prevent or treat certain types of cancer, cardiovascular disease, and neurodegenerative diseases [75,77,78].

Green tea has been found to increase the lifespan of various experimental animal models. Drinking green tea can reduce the risk of cancer, mainly because green tea is rich in catechin, especially EGCG. EGCG (25 mg/kg) extended the lifespan by diminishing the levels of inflammatory and oxidative stress in obese rats [79]. Additionally, EGCG extended the lifespan by reducing age-associated damage to liver and kidney function, improving inflammation and oxidative stress in healthy rats [80]. Green tea catechins (1 mg/kg) could also promote mice longevity by triggering mitochondrial biosynthesis [81].

Catechin can inhibit the carcinogenesis process by modulating multiple cell-signaling pathways, such as regulating the proliferation, apoptosis, angiogenesis, and killing of various types of cancer cells. One of the important hallmarks of cancer development is inflammation, and inflammatory mediators are elevated when cancer patients have a poor prognosis. Additionally, inhibition of NF-κB is a necessary step to inhibit cancer development. EGCG potently induces apoptosis and promotes cell growth arrest, by altering the expression of cell cycle regulatory proteins, activating killer caspase, and suppressing NF-κB activation [82]. In A-549 non-small lung cancer cells, EGCG exerted anticancer effects by inhibiting cell growth and inducing apoptosis, arresting cancer cell division in the G2 phase [83]. In animal and cellular experiments, EGCG dose-dependently downregulated the mRNA levels and protein expression levels of NF-κB and matrix metalloproteinase-9 in bladder cancer cells, and inhibited the cancer cell proliferation and migration in nude mice [84].

Lately, it has been indicated that EGCG might induce cytoprotective autophagy in stress events [85]. EGCG can directly combine with the autophagy-related protein LC3-I to promote the formation of LC3-II and then promote autophagy and act on the PI3K/AKT pathway to affect autophagy, apoptosis, angiogenesis, and cell cycle regulation. EGCG can reduce proliferation and induce the apoptosis of pancreatic cancer cells associated with the PI3K/AKT/mTOR pathway [86]. Filippi’s research proved that EGCG played a significant role in apoptosis, and cell cycle arrest in tumor cells by inhibiting the PI3K/AKT signaling pathways and NF-κB activity [87]. EGCG-induced apoptosis in cancer cells is associated with a marked reduction in Bcl-2 and NF-κB expression. EGCG acted on breast cancer MCF7 cells to induce apoptosis by acting on targets in the p53/Bcl-2 signaling pathway [88]. EGCG inhibited the microenvironment-induced angiogenesis of colorectal cancer through the Janus kinase/signal transducer and activator of transcription 3/interleukin-8 (JAK/STAT3/IL-8) pathway and inhibited the invasion and metastasis of colorectal cancer [89].

Additionally, AMPK activation is the major mechanism for EGCG and another catechin to influence energy metabolism. During endoplasmic reticulum stress, low concentrations of EGCG lead to autophagy-dependent survival of tumor cells by balancing the mTOR-AMPK pathway, upregulation of downregulated mTOR, and downregulation of upregulated AMPK during autophagy, enhancing cell viability [85]. Activation of AMPK by EGCG analogs resulted in inhibition of cell proliferation, downregulation of the mTOR pathway, and suppression of stem cell population in human breast cancer cells [90]. EGCG extended *Caenorhabditis elegans* lifespan through mitochondria hormesis specifically, which is a mechanism that relies on the AMPK/Sirt1/FOXO pathway [91].

In addition, catechin can also reduce oxidative stress and inflammation by inhibiting the NF-κB signaling pathway, thereby preventing the development of aging-related diseases. EGCG alleviated nonalcoholic fatty liver disease (NAFLD) in rats by reducing oxidative stress and inflammation by inhibiting PI3K/AKT/FOXO1 and NF-κB pathways [92]. In an autoimmune encephalomyelitis model, EGCG exerted an antioxidant effect by inhibiting the NF-κB signaling pathway and ameliorating its neuronal damage [93].

Catechin has potent antioxidant and anti-inflammatory properties that help prevent or treat some types of cancer. It mainly induces autophagy or apoptosis by inhibiting the NF-κB signaling pathway and activating the AMPK signaling pathway, thereby achieving the purpose of increasing the lifespan of animals.

### 3.5. Lycopene

Lycopene is a bright red carotene and carotenoid pigment found in bright red fruit and vegetables, such as tomato, watermelon, eggplant, papaya, red guava, cherry, plum, and capsicum. Tomatoes and tomato-based foods account for more than 85% of all dietary sources of lycopene [94]. The content of lycopene in ripe tomatoes is 0.03–0.14 mg/g, and the content of lycopene is positively correlated with the ripeness of tomatoes. The average daily intake of lycopene is 0.5–5 mg/d, but increasing it to 20 mg/d is not harmful to the human body [95].

Lycopene has important biological functions such as antioxidation, lowering blood lipids, anticancer, and improving body immunity [96]. Plasma lycopene levels are significantly reduced during aging. Recently, a growing body of epidemiological evidence suggested that lycopene consumption was associated with decreased risk of various chronic diseases, such as cancer, diabetes, stroke, prostate cancer, and some inflammatory diseases [97]. It is useful in the reduction in atherosclerotic plaque size and arterial stiffness. The level of lycopene in plasma is negatively correlated with the thickness and damage of the carotid artery and aortic vessel wall, which can effectively prevent the formation of atherosclerosis. In a 4-week dose-response controlled clinical trial in people at high cardiovascular risk, it was found that *trans*-lycopene might reduce the risk of cardiovascular disease by reducing the concentration of important inflammatory molecules associated with atherosclerosis [98]. A meta-analysis of tomato and lycopene supplements and cardiovascular risk factors found that increasing lycopene intake had a positive effect on improving blood lipids and blood pressure [99].

Lycopene also has potential anticancer activity. Rowles et al. found that consumption of tomatoes and their products or circulating levels of lycopene were inversely associated with cancer risk, particularly breast, colon, lung, and prostate cancers [100]. Eating vegetables or fruits rich in lycopene may protect against prostate cancer to some extent [101]. Increasing dietary intake of lycopene significantly reduced the probability of malignant tumors, and in vitro experiments proved that lycopene had a significant cell proliferation inhibitory effect on a variety of cancer cells [100]. Popular and commonly consumed tomato products increase plasma and prostate lycopene concentrations in men with prostate cancer, while lycopene supplementation has favorable activity in patients with metastatic castration-resistant prostate cancer [102,103]. Furthermore, lycopene may treat central nervous system diseases such as Parkinson’s disease and Alzheimer’s disease and preserve memory in rodents [104,105]. These beneficial effects are mainly attributed to the antioxidant properties of lycopene, which reduces the production of ROS that induce oxidative stress and is closely related to the etiology, pathogenesis of disease and the aging process [106].

Lycopene can prevent cancer through several mechanisms, including regulating signal transduction, arresting the cell cycle, and inducing apoptosis, thereby preventing the metastasis, invasion, and angiogenesis of various cancer cells. The anticancer activity of lycopene relies on its ability to inhibit oncogene expression and induce proapoptotic pathways. Lycopene can inhibit cancer cell growth by posing inhibition in the cell cycle and inducing apoptosis in the cancerous cells [104,107]. Gann’s studies showed that, in addition to its antioxidant effects, lycopene played an anticancer role by inhibiting androgen or IGF-1 signaling pathways, thereby affecting cell proliferation and growth [107]. Lycopene can slow cell cycle progression by interfering with the mitotic pathway of IGF-1, thereby affecting the development of prostate and breast cancer [108]. Lycopene’s anti-inflammatory effects depend in part on oxidative stress, MAPK/ ERK signaling pathways, and apoptosis. In the study of human colon cancer cell line HT-29, lycopene interfered with PI3K/AKT and MAPK/ERK signaling pathways by inhibiting the phosphorylation of AKT and ERK1/2 [109]. This behavior inhibited the invasion of cancer cells. Lycopene may prevent gastric cancer by inhibiting p53-dependent apoptosis, balancing apoptosis, and proliferation. Lycopene extracts from different tomato-based food products also decreased Bcl-2 and increased levels of Bax, which induce the release of cytochrome c and other proapoptotic factors from mitochondria, leading to apoptosis in prostate cancer cells [110].

Lycopene (50 mg/kg) could alleviate oxidative stress-induced neuroinflammation and cognitive impairment in D-galactose-induced accelerated aging in mice by activating the nuclear factor erythroid-2 related factor 2/NF-κB (Nrf2/NF-κB) signaling pathway [111]. Lycopene suppressed the activation of NF-κB and JNK and subsequently reduced inflammation, resulting in increased apoptosis in human colorectal cancer cells [112]. In D-galactose-induced aging in mice, lycopene in combination with nicotinamide mononucleotide could prevent cognitive impairment and attenuate oxidative damage by acting on the Keap1-Nrf2 signaling pathway [113].

Lycopene is a natural anticancer substance, and its anticancer effect occurs in a dose-dependent manner. In addition, it can delay aging by inhibiting the production of oxidation products and reducing the expression of inflammatory factors.

## 4. Conclusions

Phytochemicals in dark-colored plant foods have been shown to have beneficial effects on humans and animals and can prevent various age-related diseases, such as cardiovascular diseases, cancer, diabetes, and neurodegenerative diseases. Most of these effects were confirmed to be related to their antioxidant and anti-inflammatory capacity, as well as the regulation of autophagy and apoptosis-associated signaling pathways, including PI3K/AKT/mTOR, AMPK, MAPK, and NF-κB (Figure 3). Curcumin, catechin, and anthocyanins can promote autophagy by acting on autophagy-related targets to protect nerves, liver, and pancreas islets against age-related dysfunction. These phytochemicals can induce apoptosis, which accelerates the death of cancer cells and protects the brain from oxidative damage in mice, and can reduce myocardial cell death and slow down the development of ischemic stroke by inhibiting apoptosis. Among them, curcumin, catechin, lycopene, and quercetin mainly inhibit inflammation by inhibiting the NF-κB signaling pathway, thereby suppressing the inflammatory cascade in chronic diseases. Anthocyanins exert antioxidant and anticancer effects mainly by activating IGF-1 and MAPK signaling pathways, leading to the extension of lifespan.

Although there are numerous in vitro and in vivo studies demonstrating the antiaging effects of phytochemicals, more epidemiological and clinical intervention studies are needed to confirm these claims.

## Figures and Tables

**Figure 1 ijms-23-11038-f001:**
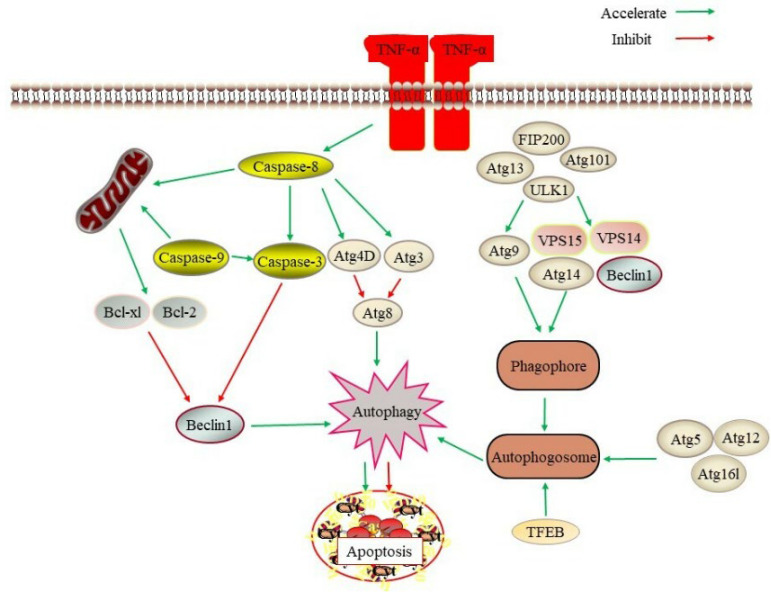
The regulation and targets of autophagy and apoptosis. ULK1, Atg13, FIP200, and Atg101 are found in a supramolecular complex that cooperates with Atg9 to promote autophagosome nucleation. ULK1 favors the autophagic cascade by facilitating a multiprotein complex formed by Beclin1, VPS34 (phosphatidylinositol-3-kinase catalytic subunit type 3), VPS15 (phosphoinositide-3-kinase regulatory subunit 4), and Atg14. A multiprotein complex composed of Atg5, Atg12, and Atg16L1 (autophagy-related 16-like 1) promotes autophagy activation. Atg3 and Atg4 are ultimately responsible for the cleavage of members of the Atg8-family proteins. Caspase-8 activates downstream targets by activating caspase-3, caspase-9, Atg3 and Atg4 to activate autophagy and apoptosis. Targeted transcription factor EB (TFEB) is a crucial regulator of the autophagy–lysosomal pathway.

**Figure 2 ijms-23-11038-f002:**
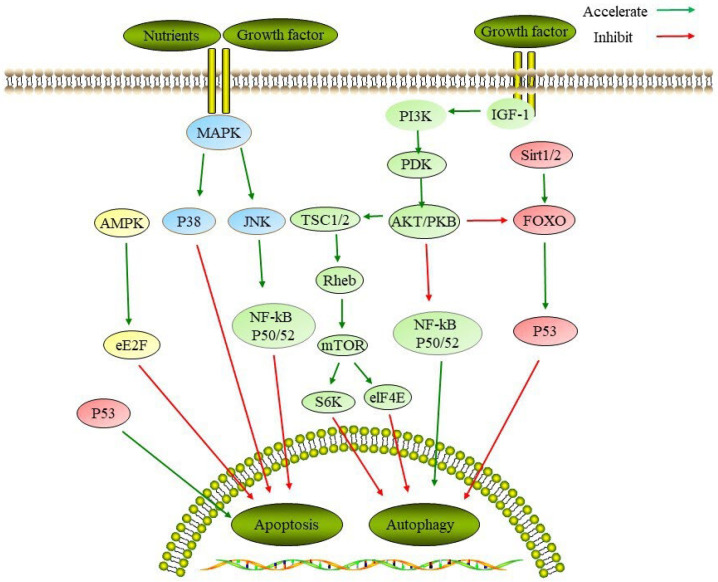
Signaling pathways regulating autophagy and apoptosis. mTOR: mammalian target of rapamycin; NF-κB: nuclear factor kappa B; AMPK: adenosine monophosphate activated; JNK: C-Jun amino-terminal kinase; S6K: ribosomal S6 kinase; eIF4E: eukaryotic initiation factor 4E; AKT: A serine/threonine kinase; PI3K: phosphatidylinositol 3-kinase; FOXO: Forkhead box o.

**Figure 3 ijms-23-11038-f003:**
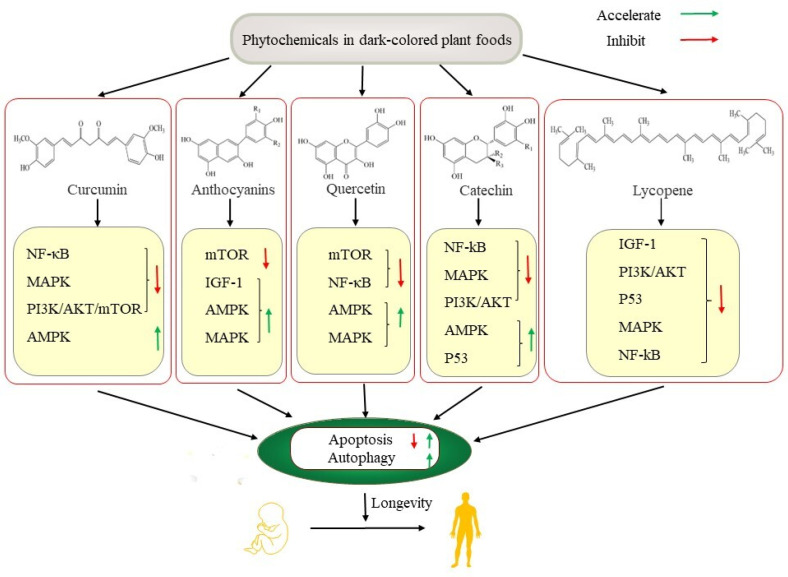
Signaling pathways of apoptosis and autophagy in the antiaging potential of phytochemicals in dark-colored plant foods. PI3K: phosphatidylinositol 3-kinase; AKT: A serine/threonine kinase; mTOR: mammalian target of rapamycin; NF-κB: nuclear factor kappa B; AMPK: adenosine monophosphate activated; MAPK: mitogen-activated protein kinase; IGF-1: insulin/insulin-like growth factor.

## Data Availability

Not applicable.

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
