# Peer review of "The Antiaging Activities of Phytochemicals in Dark-Colored Plant Foods: Involvement of the Autophagy- and Apoptosis-Associated Pathways"

_ijms, 2022, doi:10.3390/ijms231911038_

Round 1
Reviewer 1 Report
The authors summarize the recent findings on the anti-aging effects of five plant secondary metabolites in relation to autophagy and apoptosis induction. It is an excellent review, covering many recent papers.
Major concerns.
The authors use the word "pigment", which is uncomfortable as some compounds (catechins) are almost colorless. Where possible, other term such as phytochemical is preferred.
Please include mTOR, Sirt, IGF-1, etc. in Figure 1, as they appear in the text and are important for autophagy signaling.
Minor comments.
L109-111. Add reference for TFEB.
L120. comedian ?
L226. 10%-22% -> 10-22%; 0.26%-1.8% -> 0.26-1.8%
L244-246. Add reference. Ref. 63 ?
L255-256. Drosophila melano-gaster -> italic.
L263-264, 266 Cae-norhabditis elegans -> italic.
L323. (-)-epigallocatechin (EGCG) -> (-)-epigallocatechin-3-gallate (EGCG)
L352. tens in -> tensin
L441. lycopene -> Lycopene
References. Check capitalization. Italicize the scientific names.
Author Response
Point-to-point response to the reviewers’ comments:
Reviewer #1: The authors summarize the recent findings on the anti-aging effects of five plant secondary metabolites in relation to autophagy and apoptosis induction. It is an excellent review, covering many recent papers.
- Major concerns.
(1) The authors use the word "pigment", which is uncomfortable as some compounds (catechins) are almost colorless. Where possible, other term such as phytochemical is preferred.
Response: Thank you very much for taking the time to review this manuscript. According to your comments, we have made extensive modifications to our manuscript. We have rewritten the title of the manuscript to be more in line with the content of the manuscript. In addition, we have revised the manuscript content accordingly to be more in line with the theme. There are many changes in the content, please refer to the manuscript for details.
Title: The Anti-Aging Activities of Phytochemicals in Dark-Colored Plant Foods: Involvement of the Autophagy- and Apoptosis-Associated Pathways
(2) Please include mTOR, Sirt, IGF-1, etc. in Figure 1, as they appear in the text and are important for autophagy signaling.
Response: I appreciate all your comments and suggestions! We have made corrections according to your comments. Considering that adding signaling pathways in Figure 1 would make the content look cluttered, we remapped the signaling pathways in autophagy and apoptosis and placed them at the beginning of Chapter 3 (Figure 2).
- Minor comments.
(1) L109-111. Add reference for TFEB.
Response: Special thanks to you for your good comments. Based on your suggestion, we have added the references.
- Napolitano G., Ballabio A.: TFEB at a glance. J Cell Sci 2016, 129:2475-2481.
(2) L120. comedian?
Response: We are very sorry for our incorrect writing. The statements of “comedian” were corrected as “mammalian”
(3) L226. 10%-22% -> 10-22%; 0.26%-1.8% -> 0.26-1.8%
Response: We are very sorry for our incorrect writing. The statements of “10%-22%” were corrected as “10-22%”, and the statements of “0.26%-1.8%” were corrected as “0.26-1.8%”,
(4) L244-246. Add reference. Ref. 63 ?
Response: Special thanks to you for your good comments. Based on your suggestion, we have added the references.
- Dewi L.T., Adnyana M.O., Mahdi C., et al.: Study of Antocyanins Activity from Purple Sweet Potato for Reducing Apoptotic Cells Expression of The Cerebellum On Ischemic Stroke Rats. The Journal of Pure and Applied Chemistry Research 2018, 7:94-99.
(5) L255-256. Drosophila melano-gaster -> italic.
Response: Thank you very much for your comments. The statements of “Drosophila melanogaster” were corrected as “Drosophila melanogaster”
(6) L263-264, 266 Cae-norhabditis elegans -> italic.
Response: Thank you very much for your comments. The statements of “Caenorhabditis elegans” were corrected as “Caenorhabditis elegans”
(7) L323. (-)-epigallocatechin (EGCG) -> (-)-epigallocatechin-3-gallate (EGCG)
Response: We are very sorry for our incorrect writing. The statements “(-)-epigallocatechin (EGCG) -” were corrected as “(-)-epigallocatechin-3-gallate (EGCG)”
(8) L352. tens in -> tensin
Response: We are very sorry for our incorrect writing. The statements of “tens in” were corrected as “tensin”
(9) L441. lycopene -> Lycopene
Response: We are very sorry for our incorrect capitalization. The statements of “lycopene” were corrected as “Lycopene”.
- References. Check capitalization. Italicize the scientific names.
Response: Thank you very much for taking the time to review this manuscript. We have made corrections according to your comments. Because the capitalization of signaling pathways in different literature titles is different, we did not modify this. But we checked every reference and italicized the scientific names. There are many changes in the content, please refer to the manuscript for details.

Reviewer 2 Report
The manuscript is interesting and addresses the anti-ageing effects of some natural compounds based on their ability to modulate autophagy and apoptosis. The manuscript has a great potential but should be revised in terms of bibliography (up-date!) and also in order to better bring forward the main aim of the study. There are some subsections that lack the correlation with this aim. Also, the manuscript should be revised with great care to remove language discrepancies
“ There will be a funda-33 mental medical and economic issue between increasing life expectancy and the prevalence of chronic diseases [2].” – there will be? There are many countries whare this is a current issue not a future one
There is a great problem with citations used by authors – they do not match the ideas in the text.
Examples
“ Life expectancy is a significant indicator to estimate the quality of people's life. The 28 United Nations estimate global average life expectancy increased from 66.8 in 2000 to 73.3 29 in 2019, indicating the increase of the elderly population. The wasting and weakening of 30 the tissues that accompany senescence are associated with an array of age-related chronic 31 diseases, such as neurodegenerative diseases, cardiovascular diseases, chronic obstructive 32 pulmonary disease, type 2 diabetes, osteoporosis and cancers [1].”
1. Bielak-Zmijewska A.: Grabowska W., Ciolko A., et al.: The Role of Curcumin in the Modulation of Ageing. Int J Mol Sci 2019, 498 20. 499
Citation 1 refers to curcumin in ageing, not to WHO statistics or the effects of natural pigments on age related pathology; here authors should use more citations from recent literature
“ There will be a funda-33 mental medical and economic issue between increasing life expectancy and the preva-34 lence of chronic diseases [2].”
2. Janić M., Lunder M., Novaković S., Škerl P., Šabovič M.: Expression of Longevity Genes Induced by a Low-Dose Fluvastatin 500 and Valsartan Combination with the Potential to Prevent/Treat "Aging-Related Disorders". Int J Mol Sci 2019, 20.
Citation 2 does not have anything in common with the related text
Authors are advised to check the whole manuscript and use appropriate citations
“ Healthy aging and human longevity are intricate phenotypes affected by environ-39 mental factors such as physical exercise, diet, psychosocial circumstances and genetic fac-40 tors.” – genetic factors cannot be included in the category of environmental ones!
“A great number of studies on dietary intervention have illustrated that a nutritious 41 diet and caloric restriction have an essential role in healthy aging [3].”
3. Zia A., Farkhondeh T., Pourbagher-Shahri A.M., Samarghandian S.: The role of curcumin in aging and senescence: Molecular 502 mechanisms. Biomed Pharmacother 2021, 134:111119.
Since a great no of studies is mentioned, some citations should be added to support this claim, not using only one for curcumin. Just some suggestions - DOI10.1016/j.phrs.2019.104522, doi: 10.1016/j.bbadis.2019.165612., doi: 10.1159/000454924.
“ In this review, we focused on the relationship between natural edible pigments and 70 the aging process. By searching relevant literature on PubMed, Science Direct, Web of 71 Science and Embase, we comprehensively reviewed the literature about anti-aging effects 72 on plant natural edible pigments in recent years and summarized the involved anti-aging 73 signaling pathways.” – authors should provide the merhodology of the search. What ewere the key words used? Was PRISMA used or other model?
“ Targeted transcription factor EB (TFEB) is a crucial regulator of the autophagy-lyso-109 somal pathway.” – since TFEB is so important is should be included in figure 1 ?
Chapter 2.1. Autophagy is focused on the generalities mentioned in literature for autophagy and very little mentions regarding the implications of autophagy in ageing are found. This section of the manuscript should be rewritten
In section 3.1. Curcumin of the manuscript, a figure illustrating clearly the ability of curcumin to modify the pathways of autophagy/apoptosis would be useful. Also, this section should be re-organized to clearly describe the ability of curcumin to act as an anti-ageing agent by modifying apoptosis/autophagy. In the present form, this section is a collection of cellular pathways modified by curcumin, but no correlation to the aim of the study is obvious
“It alters the values of various blood pressure regulators such as vascular com-298 pliance, peripheral vascular resistance and total blood volume through its anti-inflamma-299 tory and antioxidant effects [71].” – the specific effects induced by quercetin on the membrane fluidity of cells involved in CVD development should be also mentioned - DOI10.3390/ijms13044839, DOI10.1016/j.fct.2013.02.046
“ As a ubiquitous flavonoid, quercetin is a safe dietary supplement due to its broad 318 biological effects in animals. It improves atherosclerosis and inhibits cancer development 319 by regulating IGF-1 and AMPK signaling pathways.” – results obtained in human studies should also be mentioned. The subsection regarding Q is also lacking a clear connection to the title/aim of the manuscript. It addresses the general properties of Q without stressing on data concerning the link between ageing-autophagy and apoptosis
“Natural edible pigments are a kind of food additive,…” – they are not additives!
Some data regarding the intake for the discussed products are included but they refer mainly to Asia/China. Data regarding the intake in EU countries should also be included
Some data regarding the dosage recommended for the discussed compounds should be declared in correlation with their effects on human ageing
Language corrections:
“Autophagy is a catabolism process through utilizes lysosomes to degrade damaged, 89 denatured, aged proteins and organelles in cells, which involves the sequestration and 90 transport of intracellular material to the lysosomes for degradation.” – should be rephrased for language purposes
…” autophagy can get rid of the 106 elimination of cumbersome or damaged organelles,…” – meaning???
“ It is principally executed by a family of proteases known as 130 the caspases, which are central to both the initiators and the executors of the mechanism 131 of apoptosis as they are programed cell death [18].”
“Aging is not an independent process, which often accompanied by multiple organs 170 and tissue function decline or weakened, especially in the nervous system and cardiovas-171 cular system.”
“Research on Alzheimer's disease and Parkinson's disease has also 248 shown that enzyme-mediated functional downregulation of autophagy [61].”
“They can achieve anti-oxidation,…”???
“Quercetin continuous protection 292 not only against diverse illnesses such as osteoporosis,…”???
“In addition to imparts bright color to foods, natural 453 pigments are also important for reducing the risk of age-related diseases.”???
Author Response
Point-to-point response to the reviewers’ comments:
Reviewer #2: The manuscript is interesting and addresses the anti-ageing effects of some natural compounds based on their ability to modulate autophagy and apoptosis. The manuscript has a great potential but should be revised in terms of bibliography (up-date!) and also in order to better bring forward the main aim of the study. There are some subsections that lack the correlation with this aim. Also, the manuscript should be revised with great care to remove language discrepancies
1.“ There will be a fundamental medical and economic issue between increasing life expectancy and the prevalence of chronic diseases [2].” – there will be? There are many countries whare this is a current issue not a future one
Response: It is really true as you suggested that this is a current issue not a future one. Considering that this part of the narrative involves economics, but does not match the manuscript content, we have deleted this sentence.
- There is a great problem with citations used by authors – they do not match the ideas in the text.
(1)“ Life expectancy is a significant indicator to estimate the quality of people's life. The 28 United Nations estimate global average life expectancy increased from 66.8 in 2000 to 73.3 29 in 2019, indicating the increase of the elderly population. The wasting and weakening of 30 the tissues that accompany senescence are associated with an array of age-related chronic 31 diseases, such as neurodegenerative diseases, cardiovascular diseases, chronic obstructive 32 pulmonary disease, type 2 diabetes, osteoporosis and cancers [1].”
[1] Bielak-Zmijewska A.: Grabowska W., Ciolko A., et al.: The Role of Curcumin in the Modulation of Ageing. Int J Mol Sci 2019, 498 20. 499
Citation 1 refers to curcumin in ageing, not to WHO statistics or the effects of natural pigments on age related pathology; here authors should use more citations from recent literature
Response: Thank you very much for your valuable comments. The life expectancy data in the manuscript comes from the WHO official website and does not use reference data. We have added more references to aging-related diseases to this part.
- Saheera S., Krishnamurthy P.: Cardiovascular Changes Associated with Hypertensive Heart Disease and Aging. Cell Transplant 2020, 29:963689720920830.
- Halim M., Halim A.: The effects of inflammation, aging and oxidative stress on the pathogenesis of diabetes mellitus (type 2 diabetes). Diabetes Metab Syndr 2019, 13:1165-1172.
- Yin Y., Wang Z.: ApoE and Neurodegenerative Diseases in Aging. Adv Exp Med Biol 2018, 1086:77-92.
(2)“ There will be a funda-33 mental medical and economic issue between increasing life expectancy and the preva-34 lence of chronic diseases [2].”
[2] Janić M., Lunder M., Novaković S., Škerl P., Šabovič M.: Expression of Longevity Genes Induced by a Low-Dose Fluvastatin 500 and Valsartan Combination with the Potential to Prevent/Treat "Aging-Related Disorders". Int J Mol Sci 2019, 20.
Citation 2 does not have anything in common with the related text
Response: Thank you very much for your valuable comments. Considering that this part of the narrative involves economics, but does not match the manuscript content, we have deleted this sentence.
(3) Authors are advised to check the whole manuscript and use appropriate citations
Response: Thank you very much for your valuable comments. We have made corrections according to your comments. We have appropriately deleted and supplemented some references according to the content of the manuscript and modified the format of references. There are many changes in the content, please refer to the manuscript for details.
(4) “ Healthy aging and human longevity are intricate phenotypes affected by environ-39 mental factors such as physical exercise, diet, psychosocial circumstances and genetic fac-40 tors.” – genetic factors cannot be included in the category of environmental ones!
Response: We are very sorry for our negligence in this. We've deleted the part about genetic factors.
(5) “A great number of studies on dietary intervention have illustrated that a nutritious 41 diet and caloric restriction have an essential role in healthy aging [3].”
[3]Zia A., Farkhondeh T., Pourbagher-Shahri A.M., Samarghandian S.: The role of curcumin in aging and senescence: Molecular 502 mechanisms. Biomed Pharmacother 2021, 134:111119.
Since a great no of studies is mentioned, some citations should be added to support this claim, not using only one for curcumin. Just some suggestions - DOI10.1016/j.phrs.2019.104522, doi: 10.1016/j.bbadis.2019.165612., doi: 10.1159/000454924.
Response: Thank you very much for your valuable advice, based on which we have added several references related to a nutritious diet, caloric restriction and aging.
- Zia A., Farkhondeh T., Pourbagher-Shahri A.M., Samarghandian S.: The role of curcumin in aging and senescence: Molecular mechanisms. Biomed Pharmacother 2021, 134:111119.
- Yeung S.S.Y., Kwan M., Woo J.: Healthy Diet for Healthy Aging. Nutrients 2021, 13:4310.
- Lee C., Longo V.: Dietary restriction with and without caloric restriction for healthy aging. F1000Res 2016, 5:117.
(6) “ In this review, we focused on the relationship between natural edible pigments and 70 the aging process. By searching relevant literature on PubMed, Science Direct, Web of 71 Science and Embase, we comprehensively reviewed the literature about anti-aging effects 72 on plant natural edible pigments in recent years and summarized the involved anti-aging 73 signaling pathways.” – authors should provide the merhodology of the search. What ewere the key words used? Was PRISMA used or other model?
Response: Thank you very much for your valuable advice. We have added keywords and literature databases such as “Experimental studies in vivo or in vitro, meta-analysis, randomized controlled trial and review between January 2001 and August 2022 were sought in PubMed and Science Direct, using the following search terms: ('phytochemicals' or 'autophagy' or 'apoptosis') and ('life expectancy' or 'healthy aging')”.
(6)“ Targeted transcription factor EB (TFEB) is a crucial regulator of the autophagy-lyso-109 somal pathway.” – since TFEB is so important is should be included in figure 1 ?
Response: Thank you very much for your valuable advice. We have added TEFB to Figure 1.
(7) Chapter 2.1. Autophagy is focused on the generalities mentioned in literature for autophagy and very little mentions regarding the implications of autophagy in ageing are found. This section of the manuscript should be rewritten
Response: It is true as you suggested that this section of the manuscript should be rewritten. Because my language is inappropriate, this part of the content may be a little confusing, we have re-adjusted the word order and added the corresponding content. There are many changes in the content, please refer to the manuscript for details.
(8) In section 3.1. Curcumin of the manuscript, a figure illustrating clearly the ability of curcumin to modify the pathways of autophagy/apoptosis would be useful. Also, this section should be reorganized to clearly describe the ability of curcumin to act as an anti-aging agent by modifying apoptosis/autophagy. In the present form, this section is a collection of cellular pathways modified by curcumin, but no correlation to the aim of the study is obvious
Response: Thank you very much for your comments and suggestions. We are sorry for not being able to highlight the link between curcumin and aging in the early manuscript. In the revised manuscript, we have added more explanations to the link between curcumin and aging. In the conclusion section, we summarize the related signaling pathways of phytochemicals related to autophagy and apoptosis, including curcumin (Figure 3). We did not redraw the new supplementary picture for curcumin.
There are many changes in the content, please refer to the manuscript for details.
(9)“It alters the values of various blood pressure regulators such as vascular com-298 pliance, peripheral vascular resistance and total blood volume through its anti-inflamma-299 tory and antioxidant effects [71].” – the specific effects induced by quercetin on the membrane fluidity of cells involved in CVD development should be also mentioned - DOI10.3390/ijms13044839, DOI10.1016/j.fct.2013.02.046
Response: Thank you very much for your comments and suggestions. Based on your suggestion, we have supplemented with the membrane fluidity of cells involved in CVD development on quercetin.
“Quercetin restored the transmembrane potential altered by advanced glycation end products (AGEs) in vitro, thus showing anti-diabetic potential in the late stages of diabetes [65]”
[65] Margina D., Gradinaru D., Manda G., Neagoe I., Ilie M.: Membranar effects exerted in vitro by polyphenols - quercetin, epigallocatechin gallate and curcumin - on HUVEC and Jurkat cells, relevant for diabetes mellitus. Food Chem Toxicol 2013, 61:86-93.
(10)“ As a ubiquitous flavonoid, quercetin is a safe dietary supplement due to its broad 318 biological effects in animals. It improves atherosclerosis and inhibits cancer development 319 by regulating IGF-1 and AMPK signaling pathways.” – results obtained in human studies should also be mentioned. The subsection regarding Q is also lacking a clear connection to the title/aim of the manuscript. It addresses the general properties of Q without stressing on data concerning the link between ageing-autophagy and apoptosis
Response: Thank you very much for your comments and suggestions. Based on your suggestion, we have added relevant content and literature on age-related diseases, quercetin and autophagy apoptosis. There are many changes in the content, please refer to the manuscript for details.
(11)“Natural edible pigments are a kind of food additive,…” – they are not additives!
Response: We are very sorry for our negligence in this. Based on your suggestion, we have deleted this part of the content.
- Some data regarding the intake for the discussed products are included but they refer mainly to Asia/China. Data regarding the intake in EU countries should also be included
Response: Thank you very much for your comments and suggestions. Based on your suggestion, we have added relevant content and literature.
“The average and median intakes of quercetin in Japan are 16.2-15.5 mg/d. The intake of quercetin is about 9.75 mg/d for U.S. adults [63].”
[63] Li Y., Yao J., Han C., et al.: Quercetin, Inflammation and Immunity. Nutrients 2016, 8:167.
- Some data regarding the dosage recommended for the discussed compounds should be declared in correlation with their effects on human ageing
Response: Thank you very much for your comments and suggestions. Based on your suggestion, we have added relevant content.
- Curcumin supplementation (80 mg/d) in healthy middle-aged people (40-60 years old) delayed cellular aging, reduced aging-related oxidative stress, and had a preventive effect on age-related decline in vascular function [32]
- In a clinical trial evaluating the efficacy of curcumin in delaying the development of T2DM in people with prediabetes, no one in the curcumin-treated group (1.5 g/d) was diagnosed with T2DM after 9 months of treatment [34].
- When quercetin was administered to rats at 50 mg/kg body weight, it attenuated neuronal autophagy and apoptosis in a rat model of traumatic brain injury by activating the PI3K/Akt signaling pathway [70].
- EGCG (25 mg/kg ) extended the lifespan by diminishing the levels of inflammatory and oxidative stress in obese rats [79].
- Green tea catechins (1 mg/kg) could also promote mice longevity by triggering mitochondrial biosynthesis [81].
- Language corrections:
(1)“Autophagy is a catabolism process through utilizes lysosomes to degrade damaged, 89 denatured, aged proteins and organelles in cells, which involves the sequestration and 90 transport of intracellular material to the lysosomes for degradation.” – should be rephrased for language purposes
Response: Thank you very much for your comments and suggestions. Based on your suggestion, we have revised this paragraph to better convey the theme of the manuscript.
“Autophagy is a catabolism process through utilizes lysosomes to degrade damaged, denatured, aged proteins and organelles in cells. It involves the sequestration and transport of intracellular material to the lysosomes for degradation.”
(2)…” autophagy can get rid of the 106 elimination of cumbersome or damaged organelles,…” – meaning???
Response: Thank you very much for your comments and suggestions. Based on your suggestion, we have revised this paragraph to better convey the theme of the manuscript.
“Moreover, as a cellular protection mechanism, autophagy can eliminate the damaged organelles, but overactive autophagy can also give rise to the death of cells [16, 22].”
(3)“ It is principally executed by a family of proteases known as 130 the caspases, which are central to both the initiators and the executors of the mechanism 131 of apoptosis as they are programed cell death [18].”
Response: Thank you very much for your comments and suggestions. Based on your suggestion, we have read the manuscript carefully, and for the sake of consistency of content, we have deleted this section.
(4)“Aging is not an independent process, which often accompanied by multiple organs 170 and tissue function decline or weakened, especially in the nervous system and cardiovas-171 cular system.”
Response: Thank you very much for your comments and suggestions. Based on your suggestion, we have revised this paragraph to better convey the theme of the manuscript.
“Aging is often accompanied by a decline or diminished function of multiple organs and tissues, especially the nervous and cardiovascular systems.”
(5)“Research on Alzheimer's disease and Parkinson's disease has also 248 shown that enzyme-mediated functional downregulation of autophagy [61].”
Response: Thank you very much for your comments and suggestions. Based on your suggestion, we have read the manuscript carefully, and for the sake of consistency of content, we have deleted this section.
(6)“They can achieve anti-oxidation,…”???
Response: Thank you very much for your comments and suggestions. Based on your suggestion, we have read the manuscript carefully, and for the sake of consistency of content, we have deleted this section.
” Anthocyanins are safe and with low to no toxicity, and have certain nutritional and pharmacological effects. Recent evidence suggests that anthocyanins with the capacity to modulate autophagy and apoptosis signaling pathways in aging-related diseases.”
(7)“Quercetin continuous protection 292 not only against diverse illnesses such as osteoporosis,…”???
Response: Thank you very much for your comments and suggestions. Based on your suggestion, we have read the manuscript carefully, and for the sake of consistency of content, we have deleted this section.
(8)“In addition to imparts bright color to foods, natural 453 pigments are also important for reducing the risk of age-related diseases.”???
Response: Thank you very much for your comments and suggestions. Because our title has been revised, the conclusion section has been deleted.
Other changes:
We have revised the tense of the text in the manuscript and revised the word order of some sentences. There are many changes in the content, please refer to the manuscript for details.

Round 2
Reviewer 2 Report
Authors reviewd the manuscript taking into account the previous suggestions